

# Identification, classification, and expression profile analysis of heat shock transcription factor gene family in *Salvia miltiorrhiza*

Rui Liu, Peijin Zou, Zhu-Yun Yan and Xin Chen

School of Pharmacy, Chengdu University of Traditional Chinese Medicine, Chengdu, Sichuan, China
Key Laboratory of Characteristic Chinese Medicinal Resources in Southwest, Chengdu, Sichuan, China

## ABSTRACT

In response to abiotic stresses, transcription factors are essential. Heat shock transcription factors (HSFs), which control gene expression, serve as essential regulators of plant growth, development, and stress response. As a model medicinal plant, *Salvia miltiorrhiza* is a crucial component in the treatment of cardiovascular illnesses. But throughout its growth cycle, *S.miltiorrhiza* is exposed to a series of abiotic challenges, including heat and drought. In this study, 35 HSF genes were identified based on genome sequencing of *Salvia miltiorrhiza* utilizing bioinformatics techniques. Additionally, 35 genes were classified into three groups by phylogeny and gene structural analysis, comprising 22 HSFA, 11 HSFB, and two HSFC. The distribution and sequence analysis of motif showed that SmHSFs were relatively conservative. In *SmHSF* genes, analysis of the promoter region revealed the presence of many cis-acting elements linked to stress, hormones, and growth and development, suggesting that these factors have regulatory roles. The majority of *SmHSFs* were expressed in response to heat and drought stress, according to combined transcriptome and real-time quantitative PCR (qRT-PCR) analyses. In conclusion, this study looked at the *SmHSF* gene family using genome-wide identification, evolutionary analysis, sequence characterization, and expression analysis. This research serves as a foundation for further investigations into the role of HSF genes and their molecular mechanisms in plant stress responses.

## INTRODUCTION

Plants are vulnerable to various stresses as a result of global warming, such as drought and heat (*Lesk, Rowhani & Ramankutty, 2016*; *Yao et al., 2021*). The intrinsic stress resistance systems allow plants to adapt continually in the face of various challenges, creating a complex web of stress responses (*Chen & Zhu, 2004*). Moreover, the crucial loop at the heart of this response network is transcription factors. In addition to these vast families of NAC, AP2/ERF, WRKY, and MYB (*Ambawat et al., 2013*; *Jiang et al., 2017*; *Mizoi, Shinozaki & Yamaguchi-Shinozaki, 2012*; *Nakashima et al., 2012*), a family of transcription factors called heat shock factors (HSFs) is known to have a role in plants' response to stimuli (*Guo et al., 2016*). HSFs are part of intricate signaling networks that regulate reactions to

Corresponding author
Xin Chen, chenxin@cdutcm.edu.cn

a range of abiotic stresses, including cold, high temperatures, drought, hypoxia, and soil salinity (*Andrási, Pettkó-Szandtner & Szabados, 2021*).

Previous research has demonstrated that the HSF proteins in plants are made up of five motifs, the most conserved of which is an N-terminal DNA binding domain (DBD) (*Sakurai & Enoki, 2010*; *Scharf et al., 2012*), which is connected by a flexible linker to a bipartite heptad mode oligomerization domain (OD) with hydrophobic amino acid residues (HR-A/B region) (*Baniwal et al., 2004*). The other three are a nuclear localization signal (NLS), a nuclear export signal (NES), and an activation peptide motif (AHA) (*Baniwal et al., 2004*; *Döring et al., 2000*; *Guo et al., 2016*). In accordance with the number of amino acids inserted into the HR-A/B region and the length of the flexible linker region between the DBD and the HR-A/B region, plant HSFs are divided into three classes: hSFA, HSFB, and HSFC (*Kotak et al., 2004*). Although the HR-A/B region of HSFB is small and identical to that of all non-plant HSFs, members of HSFA and class HSFC insert 21 (HSFAs) and 7 (HSFCs) with an extended HR-A/B region that contains amino acid residues situated between the HR-A and HR-B groups, respectively (*Nover et al., 1996*; *Scharf et al., 2012*). Notably, AHA motifs, which stimulate the transcription of heat shock proteins (HSPs) by binding to transcription protein complexes, are specifically found in class A members (*Scharf et al., 2012*).

The first HSF gene of the plant was cloned from tomato (*Scharf et al., 1990*). The HSF family of many more plants has been thoroughly investigated as genome sequencing efforts have progressed, including the model plant Arabidopsis, the crops rice and maize, as well as the fruits pear and pineapple (*Busch, Wunderlich & Schöffl, 2005*; *Chauhan et al., 2011*; *Lin et al., 2011*; *Qiao et al., 2015*; *Wang et al., 2021*). Heat stress controls most plant HSFs. After being exposed to heat stress, it was discovered that the transcript levels of the A2 and A6 HSFs were predominant in wheat. During drought and salt stress, numerous *TaHSFA* members, as well as B1, C1, and C2 members were also upregulated (*Xue et al., 2014*). Furthermore, it has been demonstrated that a variety of additional abiotic stressors, including heat, salt, and drought, as well as phytohormones including jasmonic acid (Ja), abscisic acid (ABA), salicylic acid (SA), and ethylene (ET), regulate plant HSF genes (*Hu et al., 2015*). Sesame has been shown to contain 30 genes that encode HSF domains. 90% of HSFs were identified to respond to drought stress by time course expression profiling. After working with several HSFA genes, it was inferred that class B-HSFs might be a significant regulator of drought response in Sesame (*Dossa, Diouf & Cissé, 2016*). Recently, 30 HSF transcription factors (*PvHSF* 1-30) were discovered in Phaseolus vulgaris. Co linearity analysis revealed that *PvHSFs* were involved in controlling the response to abiotic stresses, with the majority of *PvHSFs* differentially expressed under cold, heat, salt, and heavy metal stresses, indicating that *PvHSFs* may have various roles depending on the type of abiotic stress (*Zhang et al., 2022*).

*Salvia miltiorrhiza (Sm),* a model medical plant whose dried roots and rhizomes as medicinal parts are known as Danshen (*Lu et al., 2020*), is widely used in Asia to treat cardiovascular disease (*Li, Xu & Liu, 2018*), and Alzheimer's disease (*Zhang et al., 2016*). Water-soluble phenolic acids and lipid-soluble diterpenoid tanshinones are the distinct secondary metabolites, and two classes into which the principal active ingredients of

*S.miltiorrhiza* can be subdivided (*Jiang, Gao & Huang, 2019*). The diterpenoid compound tanshinones, which comprises tanshinone I, tanshinone II, cryptotanshinone, and dihydrotanshinone, among others, has been isolated in more than 40 various forms from S. miltiorrhiza (*Ma et al., 2015*). It has considerable anti-inflammatory, antitumor, and anti-antioxidant properties and has also been shown to have strong anticancer properties both *in vitro* and *in vivo* (*Chen et al., 2014*; *Wang et al., 2020*). Caffeic acid, salvianolic acids, rosmarinic acid (RA), and lithospermic acids are examples of substances that exhibit phenolic acid properties (*Ma et al., 2015*). These compounds have antioxidant, anticoagulant, and cell protective properties (*Wang, Morris-Natschke & Lee, 2007*). As the climate changes, the yield of *Salvia miltiorrhiza* is threatened by several concurrent stresses, including drought and high temperatures. To further produce *S.miltiorrhiza* cultivars with greater heat and drought resistance, it is crucial to understand how *S.miltiorrhiza* tolerates heat and drought.

Although it has been demonstrated that members of the HSF family respond to heat and drought stress, it is still unclear how they express themselves in response to heat stress and what molecular mechanisms drought stress in *Salvia miltiorrhiza* involves. In light of this, we discovered HSF family members from *Salvia miltiorrhiza*'s genome data and examined the expression of HSF in *S.miltiorrhiza* under drought stress based on transcriptome data. After that, *Salvia miltiorrhiza* seedlings were treated by heat stress and observed 35 *SmHSF* members' expression levels. Additionally, we combined a bioinformatic study of *SmHSFs*, including an investigation of the gene structure, motif, phylogeny, and promoter. So, in addition to analyzing the HSF family in Salvia *miltiorrhiza*, this work also assesses the expression profile of this family under heat stress and drought stress, which offers a foundation and concepts for further research into the functions of specific members.

## MATERIALS & METHODS

### Identification of SmHSF members

Genome Warehouse, BIG Data Center, project number PRJCA003150, which is available at https://ngdc.cncb.ac.cn/bioproject/browse/PRJCA003150, provided the assembly and annotation data for *Salvia miltiorrhiza* Bunge (Lamiaceae) (*Song et al., 2020*). The *S.miltiorrhiza* genome data was screened for HSF genes using the hmmsearch function in HMMER 3.0 (*Finn, Clements & Eddy, 2011*), and HSF members were identified using the hidden Markov model (HMM) corresponding to the HSF domain (PF00447) that was downloaded from the PFAM database (https://pfam.xfam.org/). *E* value is under 0.005 and default parameters were used. The *S.miltiorrhiza* genome database was searched using the BLASTP tool to find similar sequences using the *Arabidosis* HSF amino acid sequences that were obtained from TAIR (http://www.arabidopsis.org). Additionally, the SMART (*Letunic & Bork, 2018*) and CDD programs (https://www.ncbi.nlm.nih.gov/cdd/) were used to detect DBD domains in all acquired *SmHSF* proteins. Using TBtools (*Chen et al., 2020*) and the physical locational data from the *S.miltiorrhiza* genome, all of the SmHSFs were mapped to the eight chromosomes and three scaffolds of *S.miltiorrhiza* (*Zhang et al., 2021*). In addition, the ExPasy (https://web.expasy.org/compute_pi/) program

was used to explore the physical characteristics of predicted HSF proteins. Subcellular localization predictions were generated using Plant-mPLoc with default parameters (http://www.csbio.sjtu.edu.cn/bioinf/plant-multi/) (*Chou & Shen, 2010*).

### Structural, motif, and phylogenetic analysis of SmHSF genes

By comparing predicted coding sequences, the exon-intron distribution of each *SmHSF* gene was drawn using TBtools (*Chen et al., 2020*). The online tool MEME5.4.1 (https://meme-suite.org/meme/tools/meme) was used to investigate the conserved motifs of the *SmHSF* protein sequences. The number of motifs being requested is 10, and the range for motif width is 6 to 50 (inclusive). The results were visualized using TBtools. Using ClustalW in MEGAX with default parameters (https://www.megasoftware.net/), multiple sequence alignments of HSF proteins from *Arabidopsis thaliana*, *Oryza sativa* (*Chauhan et al., 2011*), *Solanum lycopersicum* (*Jin et al., 2020*), and *S. miltiorrhiza* were carried out. The sequences used for alignment are listed in Table S1. The alignment results were used to construct a phylogenetic tree using the Maximum Likelihood method with 1000 bootstrap replicates. Additionally, the evolutionary tree was embellished using Evolview (https://www.evolgenius.info/evolview/) (*Subramanian et al., 2019*).

### Cis-acting elements analysis of *SmHSFs*

The *S. miltiorrhiza* genome database provided the 2,000-bp sequence upstream and 200-bp sequence downstream from the transcription start site of each *SmHSF* gene. These sequences were used to find cis-acting regulatory elements with the online tool PlantCARE (https://bioinformatics.psb.ugent.be/webtools/plantcare/html/). A number of significant cis-acting elements were counted for each *SmHSF* using TBtools (*Chen et al., 2020*).

### Expression profiles of *SmHSFs* under drought stress based on transcriptome data

The transcriptomic data generated from the leaf and root of *S. miltiorrhiza* under moderate drought stress have been described previously (*Li, 2020*). TBtools was used to create a heat map, and after the data Log2(FPKM+1), row normalization was carried out, to make it easier to compare the expression trends of each *SmHSF* in leaves and roots and after drought stress.

### Plant materials and heat treatments

For heat stress treatment trials, tissue culture seedlings with the same or equivalent growth vigor were chosen. The *S. miltiorrhiza* tissue culture seedlings, which were grown for two months on MS medium, were moved to an artificial climate chamber with a constant temperature of 42 °C and exposed to heat stress continuously for 24 h, and six-time points (0 h, 1 h, 2 h, 6 h, 12 h, and 24 h) were selected for sample collection. All samples had three biological duplicates and were immediately frozen in liquid nitrogen following collection, and then stored at −80 °C for RNA extraction.

### RNA isolation, and cDNA synthesis

Total RNA from *S. miltiorrhiza* samples were isolated using the Plant Total RNA Isolation Kit Plus following the manufacturer's protocol (Foregene, Ltd., Chengdu, China). cDNA

synthesis was done by using 1 ug of the total RNA samples with RT Easy$^{TM}$ II Kit (Foregene, Ltd., Chengdu, China).

## Quantitative Real-Time PCR (qRT-PCR)

Reverse transcribed cDNA products were used as templates of quantitative Real-Time PCR (qRT-PCR). The reaction was carried out on a CFX Opus Real-time PCR system using the Real-Time PCR Easy $^{TM}$ -SYBR Green I (Foregene, Ltd., Chengdu, China) following the manufacturer's instructions. NCBI-BLAST Primer designed the primers for 35 SmHSF genes. *S. miltiorrhiza* Actin was used as an endogenous control for the normalization of expression levels of genes (*Jiang et al., 2020*). The relative expression levels were calculated using the $2^{-\Delta\Delta Ct}$ method. Data were analyzed using one-way ANOVA in GraphPad Prism 9 software (*, $P < 0.05$; **, $P < 0.01$; ***, $P < 0.001$; ****, $P < 0.0001$). To ensure reproducibility and dependability, three biological replications and three technical replications were implemented for each sample. The primers for the *SmHSFs* used for qRT-PCR analyses are listed in Table S2.

## RESULTS

### Identification and chromosomal localization of *SmHSF* gene family

A total of 35 *SmHSF* members were found in the *Salvia miltiorrhiza* genome after blastp and hmmsearch. These genes include HSF family domains and have been validated by NCBI-CD search and SMART. They were given the names *SmHSF1* to *SmHSF35* based on the gene IDs found in the newly sequenced genome. The members of the *SmHSF* gene family are listed in Table S3 along with their coding DNA and protein IDs and sequences.

The 35 *SmHSFs* protein sequences ranged in length from 185 bp to 508 bp, with *SmHSF11* having the longest and *SmHSF24* having the shortest. *SmHSF24* had a molecular weight of 21,538.39 Da, whereas *SmHSF11* had a molecular weight of 56,360.16 Da. The highest and lowest members were consistent with the protein sequence. And the range of the isoelectric points (pI) was 4.64 (*SmHSF25*) to 9.93. (*SmHSF29*). All *SmHSFs* were determined to be unstable, with the exception of *SmHSF35*, which was stable, according to the examination of the instability index (the stability of the protein in a test tube). Additionally, the GRAVY ranged from −0.92 to −0.481, while the Aliphatic Index (AI) ranged from 57.61 to 78.99. Plant-mPLoc subcellular localization predictions suggested that all the HSF proteins were located in the nucleus. All of the information above is displayed in following Table 1. Except for *SmHSF1*, *SmHSF2*, and *SmHSF3*, which were found on the scaffolds, the remaining genes were determined to be spread among eight chromosomes according to the mapping of 35 *SmHSFs* on the *Salvia miltiorrhiza* chromosome (Fig. 1). Each chromosome contains a different number of *S. miltiorrhiza* HSF genes, and the position of the genes does not indicate anything about their function.

### Phylogenetic, structural, and motif analysis of *SmHSF* genes

To investigate the phylogenetic relationship of 35 *SmHSFs*, a phylogenetic tree was constructed by combining *SmHSFs* with 21 *Arabidopsis* HSFs (*AtHSFs*), 25 rice HSFs (*OsaHSFs*), and 26 tomato HSFs (*SlyHSFs*) (Fig. 2). *SmHSFs* were separated into three

**Table 1 Protein information of SmHSFs.**

| Gene name | Protein length (aa) | Mw (Da) | pI | n.c.r | p.c.r | II | Stability | AI | GRAVY | Predicted location |
|---|---|---|---|---|---|---|---|---|---|---|
| SmHSF1 | 293 | 33682.07 | 8.43 | 47 | 49 | 44.92 | unstable | 57.61 | −0.872 | Nucleus |
| SmHSF2 | 332 | 38110.74 | 5.58 | 54 | 44 | 59.76 | unstable | 66.69 | −0.763 | Nucleus |
| SmHSF3 | 362 | 40936.75 | 4.84 | 60 | 41 | 55.29 | unstable | 70.25 | −0.665 | Nucleus |
| SmHSF4 | 290 | 31359.95 | 5.89 | 38 | 36 | 64.19 | unstable | 65 | −0.611 | Nucleus |
| SmHSF5 | 364 | 42052.88 | 5.84 | 62 | 55 | 55.31 | unstable | 76.59 | −0.707 | Nucleus |
| SmHSF6 | 341 | 37900.71 | 8.22 | 31 | 33 | 57.96 | unstable | 67.8 | −0.496 | Nucleus |
| SmHSF7 | 357 | 41644.68 | 5.72 | 60 | 52 | 61.34 | unstable | 61.23 | −0.92 | Nucleus |
| SmHSF8 | 233 | 26510.93 | 8.74 | 35 | 38 | 54.08 | unstable | 65.32 | −0.803 | Nucleus |
| SmHSF9 | 316 | 34592.59 | 4.88 | 54 | 38 | 52.21 | unstable | 78.99 | −0.481 | Nucleus |
| SmHSF10 | 345 | 37892.18 | 5.36 | 45 | 39 | 58.31 | unstable | 66.75 | −0.594 | Nucleus |
| SmHSF11 | 508 | 56360.16 | 5.26 | 67 | 49 | 61.03 | unstable | 76.38 | −0.559 | Nucleus |
| SmHSF12 | 328 | 37620.4 | 6.29 | 49 | 46 | 50.25 | unstable | 74.27 | −0.688 | Nucleus |
| SmHSF13 | 259 | 29742.92 | 9.19 | 37 | 42 | 52.13 | unstable | 77.53 | −0.725 | Nucleus |
| SmHSF14 | 482 | 53637.43 | 5.36 | 72 | 55 | 59.59 | unstable | 66.78 | −0.754 | Nucleus |
| SmHSF15 | 352 | 39746.64 | 5.92 | 46 | 38 | 56.96 | unstable | 71.7 | −0.628 | Nucleus |
| SmHSF16 | 388 | 43871.84 | 5.72 | 52 | 41 | 58.61 | unstable | 68.09 | −0.696 | Nucleus |
| SmHSF17 | 265 | 30541.47 | 6.72 | 34 | 33 | 53.25 | unstable | 71.4 | −0.685 | Nucleus |
| SmHSF18 | 252 | 28699.33 | 5.9 | 33 | 29 | 45.44 | unstable | 74.33 | −0.581 | Nucleus |
| SmHSF19 | 260 | 30162.06 | 5.77 | 35 | 32 | 48.55 | unstable | 78 | −0.662 | Nucleus |
| SmHSF20 | 187 | 21664.05 | 9.76 | 19 | 31 | 43.54 | unstable | 64.12 | −0.591 | Nucleus |
| SmHSF21 | 290 | 33416.89 | 6.92 | 33 | 32 | 63.92 | unstable | 71.24 | −0.638 | Nucleus |
| SmHSF22 | 425 | 48238.63 | 5.44 | 62 | 46 | 60.87 | unstable | 68.35 | −0.752 | Nucleus |
| SmHSF23 | 502 | 55086.51 | 4.78 | 68 | 44 | 61.74 | unstable | 62.01 | −0.627 | Nucleus |
| SmHSF24 | 185 | 21538.39 | 8.97 | 22 | 25 | 48.8 | unstable | 68.54 | −0.698 | Nucleus |
| SmHSF25 | 383 | 43547.77 | 4.64 | 65 | 39 | 47.29 | unstable | 71.31 | −0.653 | Nucleus |
| SmHSF26 | 227 | 26433.93 | 6.97 | 34 | 34 | 51.7 | unstable | 68.77 | −0.735 | Nucleus |
| SmHSF27 | 313 | 36130.43 | 5.11 | 55 | 39 | 44.08 | unstable | 66.68 | −0.792 | Nucleus |
| SmHSF28 | 399 | 45225.38 | 5.16 | 59 | 41 | 47.92 | unstable | 68.67 | −0.738 | Nucleus |
| SmHSF29 | 187 | 22125.45 | 9.93 | 21 | 35 | 62.15 | unstable | 65.24 | −0.9 | Nucleus |
| SmHSF30 | 336 | 37664.02 | 4.89 | 47 | 32 | 58.12 | stable | 66.76 | −0.505 | Nucleus |
| SmHSF31 | 353 | 39554.09 | 5.7 | 48 | 39 | 59.9 | unstable | 64.67 | −0.724 | Nucleus |
| SmHSF32 | 487 | 54256.95 | 5 | 68 | 47 | 60.98 | unstable | 66.45 | −0.575 | Nucleus |
| SmHSF33 | 341 | 39147.22 | 5.39 | 64 | 49 | 52.12 | unstable | 62.02 | −0.734 | Nucleus |
| SmHSF34 | 224 | 25709.09 | 8.32 | 32 | 34 | 51.65 | unstable | 78.75 | −0.632 | Nucleus |
| SmHSF35 | 269 | 29506 | 6.34 | 35 | 34 | 34.25 | stable | 60.26 | −0.686 | Nucleus |

**Notes.**

MW(Da), Molecular weight in Dalton; pI, isoelectric point; n.c.r, total number of negatively charged residues (Asp + Glu); p.c.r, total number of positively charged residues (Arg + Lys); II, the instability index; AI, Aliphatic index; GRAVY, Grand average of hydropathicity; Predicted location, Predicted Subcellar location.

groups (HSFA/HSFB/HSFC) in accordance with the grouping of HSFs in *Arabidopsis*. Each group has different member distributions. The distribution of members in these groups is like that in *Arabidopsis thaliana*, with Group A having the most members (22 genes) and

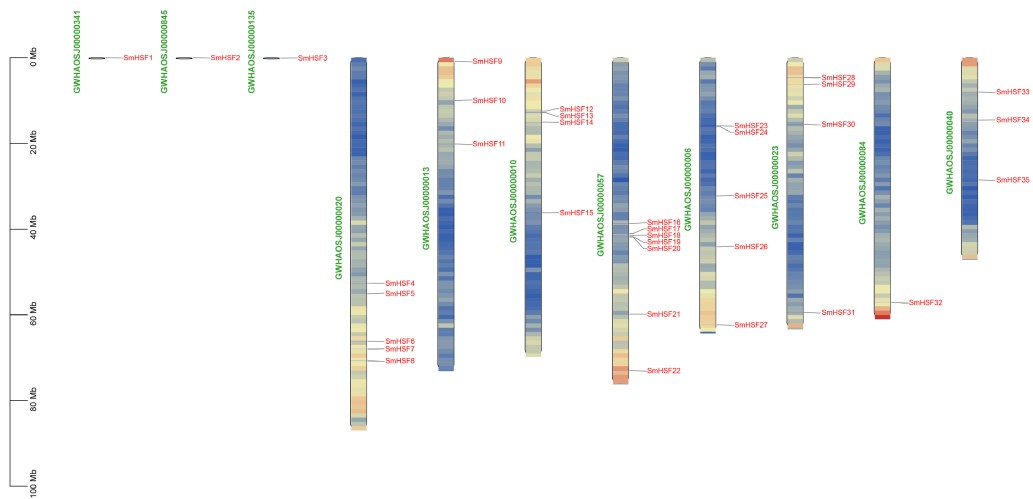

**Figure 1  Distribution of *SmHSF* genes among 8 chromosomes and 3 scaffolds.**

Group C having the fewest members (two genes). Group A of *SmHSF* consisted of nine subgroups, from A1-A9; group B, with four subgroups (B1–B4); and group C, with only one subgroup (C1).

To further examine the structural diversity of genes, we analyzed the gene structures of 35 *SmHSF* genes. Analysis revealed that the majority of *SmHSFs* members belonged to the same subgroup and shared exon and intron structures (Fig. 3B). The number of exons and introns differs among *SmHSFs*, however. Of the 35 *SmHSFs*, 32 have two exons, two genes have three exons, and interestingly, *SmHSF24* has five exon regions, two of which are incredibly short, and no UTR was discovered. All *SmHSF* genes have introns, of which 28 genes have one, five have two, one has three, and one has four.

Ten motifs were found in the *SmHSF* family members using motif analysis with MEME, the distribution of motifs corresponding to the phylogenetic tree of the *SmHSFs* family is shown in Fig. 3. Comparable motif compositions among the *SmHSFs* proteins clustered in the same subgroup imply that the members of the subgroup share similar activities. The DBD domain and HR-A/B are both present in the *SmHSF* family, just like the HSF family in other species. Motif2, motif3, and motif4 could be found in each member. Following sequence alignment, it was discovered that motif 1, motif 2, motif 3, and motif 6 make up the DBD domain of HSF, whereas motifs 4 and motifs 5 represent HR-A and HR-B, respectively. Except for *SmHSF13* and *SmHSF24*, we detected motif1 and motif6 in all members. Only group A members possess motif 5, which has not been discovered in members of groups B or C. Additionally, only a few members of group A possess motifs 7, 8, and 9, where motif 8 is AHA and motif 9 is NLS. The sequence and distribution information of all motifs are in Table S4.

## Cis-acting elements analysis of *SmHSFs*

To further learn about the biological function of *SmHSFs*, the cis-acting elements in the 2,000 bp upstream and 200 bp downstream sequences from the transcription start sites of

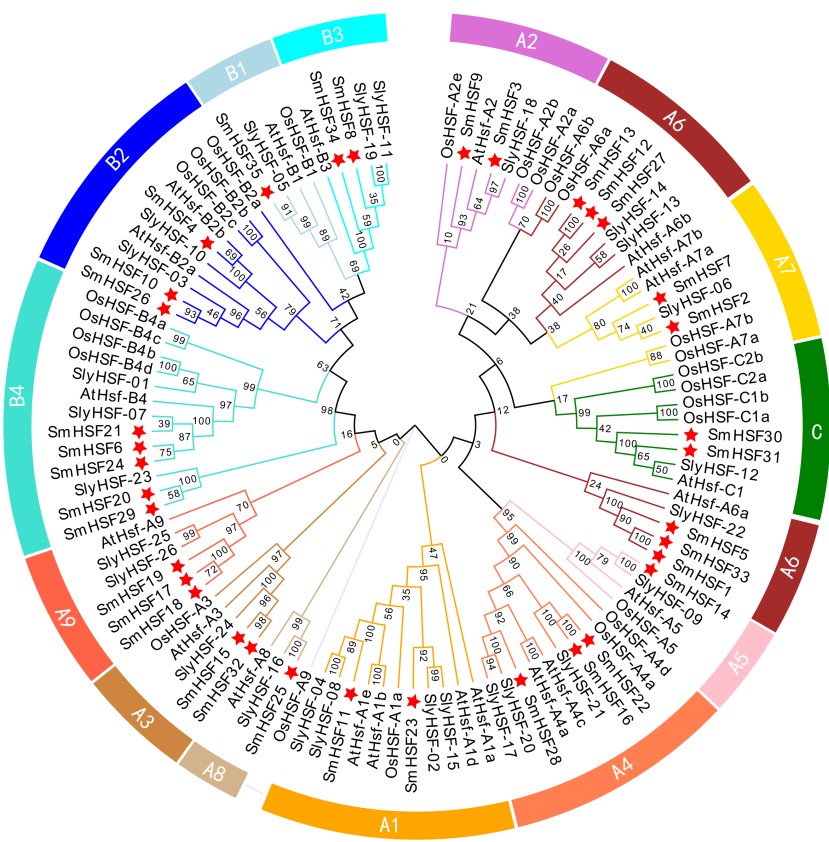

**Figure 2** Phylogenetic relationships among the HSF family members of *S. miltiorrhiza*, Arabidopsis, rice, and tomato. Full-length amino acid sequences were aligned using ClustalW and phylogenetic trees were constructed using the ML method in MEGAX. The tree clusters HSF proteins into different groups, which are represented by different colors within the branches.

*SmHSFs* were examined using PlantCARE. A variety of cis-acting components, including those related to stress, hormones, and development, constitute the promoter of each *SmHSF* (Fig. 4). Members of *SmHSFs* are broadly diversified in terms of development-related components. The ABA-responsive element (ABRE), ABA and drought responsive element (ABRE3a), methyl jasmonate (MeJA)-responsive element (CGTCA-motif), and another MeJA-responsive element (TGACG motif), the ethylene-responsive element (ERE), gibberellin-responsive element (P-box), and cis-acting element involved in salicylic acid responsiveness (TCA-element) were all found in 30, 20, 30, 30, 28, 7, and 23 *SmHSFs*, respectively. Also, several light-associated cis-acting elements, like ACE and G-Box, as well as components connected to stress. MYB element, MYC element, DRE core (drought, salt, low temperature, and ABA responses), and STRE element (activated by heat shock, osmotic stress, low pH, and nutrient starvation) were identified in the promoters of 33, 33, 10, and 24 *SmHSF* genes, respectively. Additionally, anaerobic induction elements (AREs), low-temperature responsiveness elements (LTRs), MYB binding site involved in drought-inducibility (MBSs), defense and stress responsiveness (TC-rich repeats),

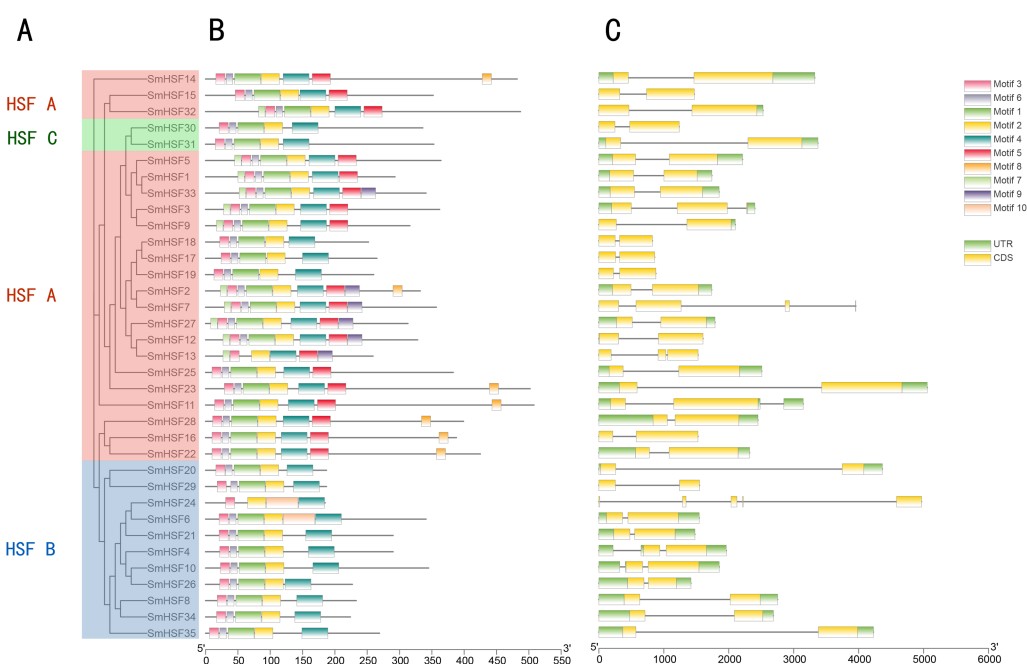

**Figure 3** **Phylogenetic relationship, conserved protein motif structure and gene structure of SmHSFs.**
(A) A phylogenetic tree was constructed based on the full-length sequence of *Salvia miltiorrhiza* HSFs protein using MEGA X software, including groups A, B, and C. (B) Motif composition of *Salvia miltiorrhiza* HSF protein. Motifs numbered 1–10 are shown in colored boxes, and the scale at the bottom indicates the length of the protein. Sequence information for each motif is in Table S4. (C) Gene structure of *Salvia miltiorrhiza* HSF gene. UTRs, untranslated regions, are represented by green boxes; CDS, coding sequences, are represented by yellow boxes. Black lines represent introns.

wounding and pathogen responsiveness element (W-box), and wound-responsive element (WUN-motif) were found in 27, 18, 18, 10, 15, and 20 *SmHSFs*, respectively. However, the heat stress responsiveness element (HSEs) is only found in *SmHSF7*. These findings show that *SmHSFs* may be associated with a variety of transcriptional regulations involving hormones, stress responses, and development.

## Expression profiles of *SmHSFs* under drought stress

Tanshinone and salvianolic acid, have been observed to accumulate in the roots and leaves of *S. miltiorrhiza*, respectively (*Sha, 2015*; *Wang & Wu, 2010*). Additionally, drought stress will have some effect on HSF. To further investigate the expression of HSF in these two tissues and the expression changes in response to drought stress, we selected previous transcriptome data for HSF expression analysis. *SmHSF23* and *SmHSF 26* were not present in the transcriptome data, while other members were expressed to different degrees in roots and leaves as well as during drought stress. There were variations in the expression patterns of *SmHSFs* in roots and leaves, as shown in Fig. 5. *SmHSF21* and *SmHSF29* were not expressed in leaves or roots, respectively. There were 11 members, with the expression in roots being lower than in leaves. Of them, *SmHSF10*, *SmHSF14*, *SmHSF16*, and *SmHSF34* clearly differed from the others. Notably, members like *SmHSF20*, *SmHSF6*, *SmHSF8*,

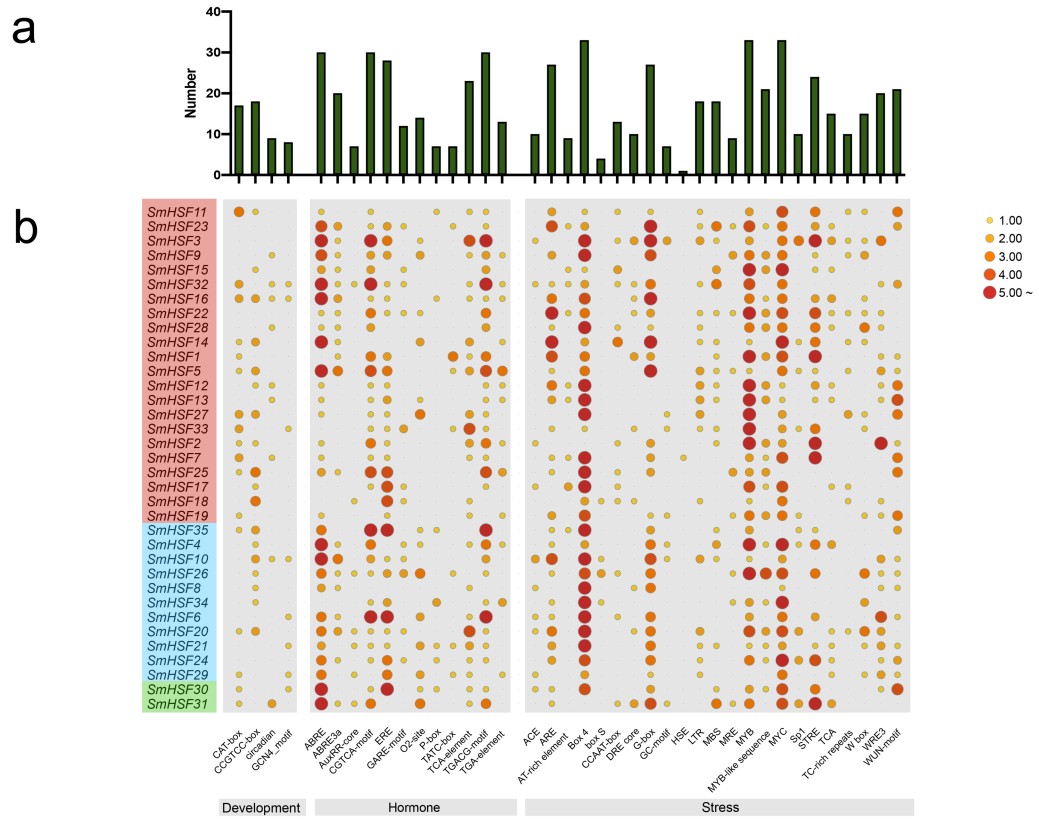

**Figure 4  Analysis of the cis-acting elements in the promoter regions of the SmHSF genes.**
(A) The graph was generated based on the presence of cis-acting element responsive to specific
processes/eleictors/conditions (*x*-axis) in HSF gene family members (*y*-axis). (B) Heatmap of the number
of cis-acting elements in each SmHSF. Based on the functional annotation, the cis-acting elements were
classified into three major classes: development-, hormone-, stress-related cis-acting elements.

*SmHSF3*, and SmHSF1 turned out to express mostly in the roots. Certain members like
*SmHSF10*, *SmHSF16*, *SmHSF30*, *SmHSF14*, and *SmHSF28* have varying expression in
the roots and leaves and generally have greater expression levels. Following treatment for
drought stress, it was found that the expression of 16 genes was up-regulated in the leaves,
with *SmHSF9* being the most visibly changed. It was discovered that the expression of 18
genes was up-regulated in roots, with *SmHSF1*, *SmHSF2*, *SmHSF3*, *SmHSF5*, and *SmHSF33*
standing out. Eleven *SmHSF* genes (*SmHSF1, 2, 8, 11, 12, 13, 15, 16, 22, 27,* and *35*) are
expressed in both roots and leaves under drought stress, and these genes may be activated
by drought stress, further studies are required on how the functions.

### Expression profiles of *SmHSFs* under heat stress

In response to heat stress, HSF genes are crucial for plant heat tolerance. To show how
HSF genes react to heat stress in our work, the expression patterns of the *SmHSFs* gene
family were identified using qRT-PCR. The results are depicted in Fig. 6. Despite barely
being expressed, *SmHSF25* rose in response to heat stress. When exposed to heat stress,
the expression of *SmHSF11*, *SmHSF14*, and *SmHSF30* did not change considerably, and

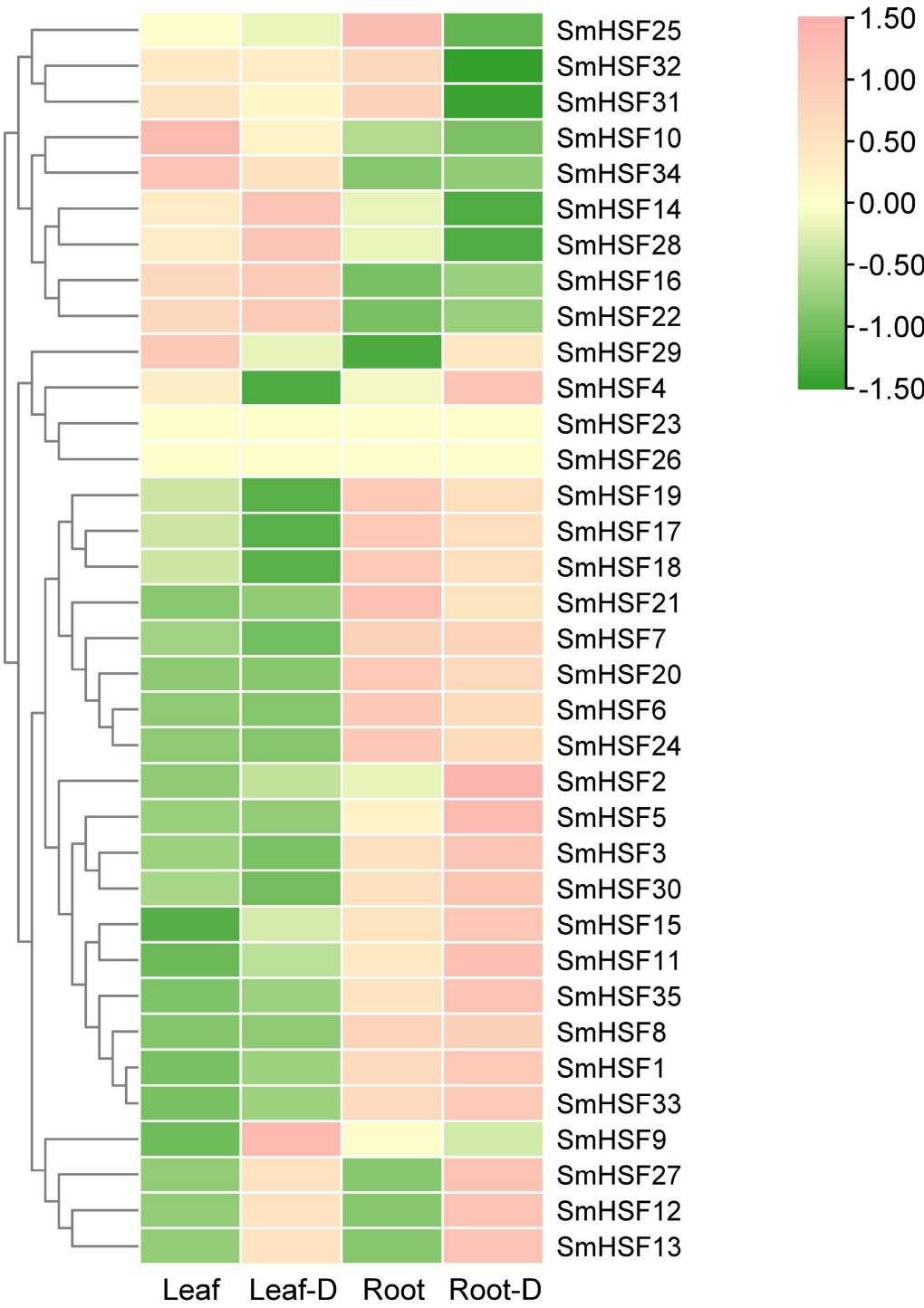

**Figure 5** **Expression pattern of SmHSFs genes in *Salvia miltiorrhiza* leaf and root under drought stress.** Log2(FPKM+1) values were used for the heatmap. D, Drought. The color scale represents single row.

the expression of SmHSF33, SmHSF34 were repressed, while the expression of other *SmHSF* genes was up-regulated to varying degrees. Notably, the expression of *SmHSF2*, *SmHSF7*, and *SmHSF9* was markedly upregulated in response to heat stress, particularly *SmHSF9*, which was almost 4000-fold greater than the control at 2 h, showing that *SmHSF2*, *SmHSF7*, and *SmHSF9* were implicated in the heat stress response pathway. In addition, the expression of *SmHSF3*, *SmHSF4, SmHSF15, SmHSF27, SmHSF31, SmHSF32,* and *SmHSF35* also changed significantly after heat stress, and these genes are all worthy of further consideration.

## DISCUSSION

Transcription factors are crucial for plant growth, development, and stress resistance. Some transcription factor families in *Salvia miltiorrhiza*, including WRKY, AP2/ERF, MYB, bHLH, and NAC (*Wu et al., 2021*; *Zhang et al., 2021*) have been found, and yet nothing is known about the HSF transcription factor family. By controlling the expression of stress-responsive genes, HSFs play a crucial role in how plants react to various abiotic stimuli (*Guo et al., 2016*). In this study, we first discovered 35 HSF genes in *Salvia miltiorrhiza* and extensively analyzed these genes.

The 35 *SmHSF* genes are relatively small in number, being greater than those found in *Arabidopsis* (21), rice (25), and strawberry (17), but lower than those found in wheat (78) and Brassica juncea (60) (*Hu et al., 2015*; *Li et al., 2020*; *Zhou et al., 2019*). The *S. miltiorrhiza* HSF gene family was further separated into groups A, B, and C with molecular phylogenetic analysis. The largest group, group A, had 22 *SmHSFs*, which were separated into nine subgroups, A1 to A9. Group B, which contained 11 HSFs, including B1 to B4, came next. Only two individuals make up Group C, and all appear to be C1 members. Six individuals made up the *Salvia miltiorrhiza* A6 group, but it's been previously demonstrated that members of this group are significantly less numerous in dicotyledonous plants than in monocotyledonous plants, indicating that this subgroup of *Salvia miltiorrhiza* may be the objective of more stringent purification selection (*Wang et al., 2018*). No members of the HSF-B5 and C2 subgroups were found in *S. miltiorrhiza*, which is following *Arabidopsis*, where C2 is a monocotyledon-specific gene that has only been detected in monocotyledonous plants (*Guo et al., 2016*; *Wang et al., 2018*). This uneven distribution of several groups demonstrates how their roles in the genome evolution of *Salvia miltiorrhiza* have changed over time.

By analyzing the structure and motif, the function of the gene was further defined. It was discovered that each subgroup had a similar distribution of the gene structure (Fig. 3), suggesting that the functions of the genes within the subgroup may be comparable. The DBD domain, which is highly conserved in plants, has about 100 amino acid residues (*Scharf et al., 2012*), as do the HSF genes in *Salvia miltiorrhiza*. On the other hand, the DBD domains of SmHSF13 and SmHSF24 only have 62 and 63 amino acid residues, respectively, indicating that their DBD domains are insufficient, which may be brought on by incomplete genome assembly. Only five members of group A had AHA motif identified, namely *SmHSF11* from the A1 subgroup, *SmHSF14* from the A5 subgroup, *SmHSF16, SmHSF22,*

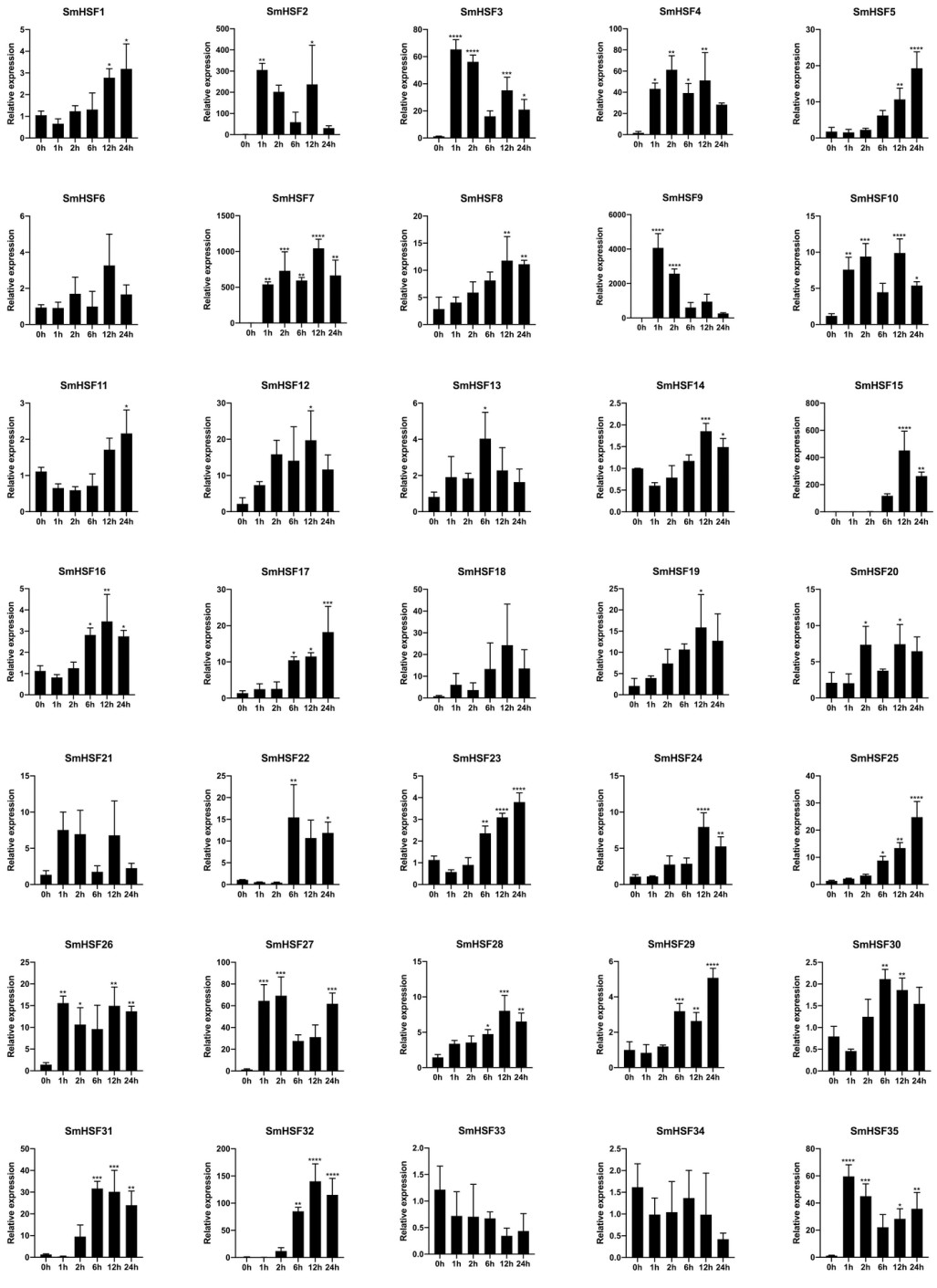

**Figure 6** **Relative expression level of SmHSFs analyzed by qRT-PCR response to heat stress treatment.** qRT-PCR data were normalized using *Salvia miltiorrhiza* Actin gene and are shown relative to 0 h. *X*-axes are time course (0 h, 1 h, 2 h, 6 h, 12 h, and 24h) and *y*-axes are scales of relative expression level. All data represent means ±SD of three independent replicates. Statistical significance was analyzed by one-way ANOVA (*, $P < 0.05$; **, $P < 0.01$; ***, $P < 0.001$; ****, $P < 0.0001$).

and *SmHSF28* from the A4 subgroup, which all contribute to activator potential (*Kotak et al., 2004*). Without the AHA domain, other members may carry out their responsibilities by combining with other A-class HSFs to create hetero-oligomers, somewhat like B- and C-class HSFs (*Guo et al., 2008a*).

The analysis of the cis-acting elements in the promoter region identified numerous light responsive elements, MYB elements, MeJA elements, ABRE elements, STRE elements, and other elements, all of which suggest that these SmHSF genes are involved in hormone-related plant growth and development as well as in response to various stresses. The expression response of SmHSFs to heat and drought stress appears to be associated, particularly with those cis-acting components relevant to abiotic stress, such as MYB, MYC, ABRE, DRE, MBS, and STRE. The medicinal plant *Salvia miltiorrhiza* contains mostly water-soluble phenolic acids and fat-soluble tanshinones, which have been shown to accumulate predominantly in the roots and leaves. Previously, the HSF family was discovered to be tissue- and stage-specific in pineapple (*Wang et al., 2021*), Carnation (*Li et al., 2019*), and Tartary buckwheat (*Liu et al., 2019*). Although there were different expression levels in the roots and leaves of *Salvia miltiorrhiza*, it was discovered from the transcriptome data that the HSF gene did not appear to have any apparent tissue specificity, this finding was in line with that of soybean and sesame (*Dossa, Diouf & Cissé, 2016*; *Lin et al., 2014*). This indicated that the function of the HSF gene family was conserved in *Salvia miltiorrhiza*. 16 genes were discovered to be up-regulated in leaves, and 18 genes were found to be up-regulated in roots following treatment for drought stress; this up-regulation behavior was previously seen in *Salix suchowensis* (*Zhang et al., 2015*), and *Chenopodium quinoa* (*Tashi et al., 2018*). Numerous cis-acting elements linked to drought were found among the 11 SmHSFs whose expression was frequently increased following drought stress, including MYB, MYC, ABRE3a, DRE, and MBS, which were demonstrated to be linked to the regulation of HSF expression during drought (*Li et al., 2019*). These genes are expected to play a significant part in how *S. miltiorrhiza* responds to drought and other abiotic challenges (*Zhu et al., 2017*).

After heat treatment, 35 *SmHSFs* were investigated for expression. Among them, during heat stress treatment, 29 genes were up-regulated and two genes were down-regulated. *SmHSF2*, *SmHSF7*, and *SmHSF9* are significantly up-regulated, particularly *SmHSF9*, which is almost 4000 times greater than the control at 2 h. The up-regulation of *SmHSF7* also reached a peak at 12 h, which was about 1,000 times the relative expression of the control group, which may be related to the only HSE cis-acting element found in this gene, which significantly activated in response to heat stress. Along with HSE, STRE is a critical element of the heat stress response (*Zhu et al., 2017*). The Arabidopsis HSFA1a direct binding site for the STRE element was also discovered (*Guo et al., 2008b*). This suggests that STRE is present in those SmHSFs whose expression is up-regulated in response to heat stress, potentially pointing to differences in the regulation of several genes. Other genes whose expression value increased more than dozens of times after heat stress, such as *SmHSF3*, *SmHSF4, SmHSF15, SmHSF27, SmHSF31, SmHSF32,* and *SmHSF35* are considered to be extremely sensitive to heat stress and play an essential role in response to heat stress. In fact, plants frequently experience many stresses along their development.

*SmHSF2, SmHSF8, SmHSF15, SmHSF27, SmHSF35*, and other genes that respond to both heat and drought stress should be the main targets, and it is worthy of further research the regulatory mechanism of *S.miltiorrhiza* in response to abiotic stress.

# CONCLUSIONS

Thirty-five *SmHSF* genes were identified from the *Salvia miltiorrhiza* genome, which were systematically examined using approaches such as chromosomal position, gene structure, motif, phylogenetic relationship, and cis-acting element analysis. RNA-seq data was used to examine the expression profile of the HSF gene in the roots and leaves of *Salvia miltiorrhiza* under drought stress. The changes in HSF under stress were evaluated in conjunction with the expression study of the HSF gene in *S.miltiorrhiza* under heat stress, which helped identify some methods that could increase *Salvia miltiorrhiza* tolerance to stress. Differences in the regulation of various genes may be explained by integrating the expression data with the findings of promoter analysis, which may also contribute to further screening of genes that are tolerant to heat and drought. The findings provide a basis for further studies on the function and molecular mechanism of HSF genes in plant stress response.

# ACKNOWLEDGEMENTS

All of you who contributed to the experiments and the article are greatly appreciated.

### Funding
This study was funded by the Natural Science Foundation of China (81973416). The funders had no role in study design, data collection and analysis, decision to publish, or preparation of the manuscript.

### Grant Disclosures
The following grant information was disclosed by the authors:
Natural Science Foundation of China:  81973416.

### Competing Interests
The authors declare there are no competing interests.

### Author Contributions
- Rui Liu conceived and designed the experiments, performed the experiments, analyzed the data, prepared figures and/or tables, authored or reviewed drafts of the article, and approved the final draft.
- Peijin Zou performed the experiments, prepared figures and/or tables, and approved the final draft.
- Zhu-Yun Yan conceived and designed the experiments, authored or reviewed drafts of the article, and approved the final draft.
- Xin Chen conceived and designed the experiments, authored or reviewed drafts of the article, and approved the final draft.

## Data Availability

The raw measurements are available in the Supplementary Files.

## Supplemental Information

Supplemental information for this article can be found online at http://dx.doi.org/10.7717/peerj.14464#supplemental-information.

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
