# Peer review of "Identification, classification, and expression profile analysis of heat shock transcription factor gene family in Salvia miltiorrhiza"

_PeerJ, doi:10.7717/peerj.14464_

## Round 0.1 · original submission · Minor Revisions

I agree with the reviewers that you need to repeat the promoter analysis using sequence with respect to the Transcription start site (2000 bp upstream and 200 bp downstream). However, you can use the method and software you have used; No need to change the software. It will be an interesting to identify, if these HSFs have any TF-target relationship with each other. This will help to further identify the closer gene-network involved in heat response.

Reviewer 1 ·

Basic reporting

Salvia miltiorrhiza, as a medicinal plant, focus on the secondary metabolites firstly. So it is inappropriate not to put forward in the foreword of the paper.

Experimental design

The experimental design is relatively common, but the analysis figure of the promoters elements is very intuitive,but some data had not to be shown effectively. Many results should be relevant, and should be highlighted in the discussion, for instance, CIS-element and genes expression result could be verified each other. The candidate genes should be picked and be further analysed.
How to show the protein stability?
How to testify HSF as the transcription factors , the subcellular localization experiment of some genes is necessary.
In Figure 2, why contains two stereotypes, A2 and A7?

Validity of the findings

The conclusions should be appropriately stated.

·

Basic reporting

The manuscript entitled “Identication, classication, and expression profile analysis of
heat shock transcription factor gene family in Salvia miltiorrhiza” by Liu et al. aimed to perform a genome-wide identification, evolutionary analysis, sequence characterization, and expression analysis of heat shock transcription factor gene family in Salvia miltiorrhiza. The research work is important, as S.miltiorrhiza is a medicinal herb. Heat and drought stress can severely affect the growth of the plant. The authors have done a detailed analysis of HSF family in Salvia miltiorrhiza.

However, I have the following major critical comments for the authros:

1. a. In the Materials and Methods, line 152, under the section Cis-acting elements analysis of SmHSFs, the authors have mentioned that “The S.miltiorrhiza genome database provided the 2000-bp sequence upstream from the initiation codon of each SmHSF gene. These sequences were used to find cis-acting regulatory elements with the online tool PlantCARE (https://bioinformatics.psb.ugent.be/webtools/plantcare/html/).
b. In the results, line 242, under the section Cis-acting elements analysis of SmHSFs, the authors have mentioned that “To further learn about the biological function of SmHSFs, the cis-acting elements in the 2 kb upstream sequences from the translation start sites of SmHSFs were examined using PlantCARE.”

Actually, the promoter sequence is always in the upstream of the Transcription Start Site (TSS). However, the authors have taken 2000 bp upstream of ATG site, which is TIS. Hence, the sequence they have taken could be a part of Exon 1 and a part of promoter or only Exon 1 depending on the gene length. Hence, the cis-element detected in those regions are not correct, as the sequence is not right. Sometime, some distal cis-elements are present in the downstream of promoter sequence, but majority are present in the upstream part of promoter sequence. Hence, please extract the promoter sequence from the upstream of TSS and perform the cis-element enrichment analysis.

The authors have used PlantCARE database for promoter analysis. However, this database is very old and does not have much information. Hence, you are certainly going to miss many important advanced cis-elements that are recently identified. The best option is to use the MATCH program in TRANSFAC (geneXplain) which you need to pay for the subscription. However, the authors can use PlantPAN and PLACE database if you do not have access to TRANSFAC. PlantPAN is quite up to date.

Hence, the authors needs to repeat this promoter anlysis by extracting the correct promoter sequences.


2. The language needs to be polished. There are many mistakes. For example:
a. First line of the abstract: “In response to abiotic stresses, transcription factors play indispensable roles.” Please rephrase it.
b. In the abstract, line 42: The gene family’s motif distribution and sequence analysis…please repharse…
c. In the Introduction, line 80: “The tomato was the plant where HSF genes were initially cloned”….please rephrase it. `
d. Line 93: of Sesame’s drought response
e. Line 100: Salvia miltiorrhiza (Sm), a model medicinal plant (Lu et al. 2020), the dried roots and rhizomes as medicinal parts known as Danshen, are used widely in Asian nations to treat cardiovascular disease (Li et al. 2018)
f. Line 114: “we gave treatments to S.miltiorrhiza seedlings”
Please check the language throughout the manuscript. T

Experimental design

Experimental design is fine except the promoter analysis part as the authors have taken wrong sequences.

Validity of the findings

The findings are interesting except the promoter analysis part.

Additional comments

The authors need to repeat the promoter cis-elemnt identification by taking correct promoter sequences.
Hence, they need to present result and discussion based on new analysis.

---

## Round 0.2 · accepted · Accept

The authors have made all the needed corrections.

·

Basic reporting

I do not have any more comments. The authors have revised the manuscript and addressed my comments.

Experimental design

The experimental design is fine.

Validity of the findings

The research work is interesting and important.